# SimbaML: Connecting Mechanistic Models and Machine Learning with Augmented Data

**Maximilian Kleissl**[1,*], **Lukas Drews**[1,*], **Benedict B. Heyder**[1,*,†], **Julian Zabbarov**[1,*]
{firstname.lastname}@student.hpi.de, [†]bjoern.heyder@student.hpi.de

**Pascal Iversen**[1], **Simon Witzke**[1], **Bernhard Y. Renard**[1,2], **Katharina Baum**[1,2,3]
{firstname.lastname}@hpi.de

[1]Hasso Plattner Institute, Digital Engineering Faculty, University of Potsdam, Germany
[2]Department of Mathematics and Computer Science, Free University Berlin, Germany
[3]Windreich Department of Artificial Intelligence and Human Health & Hasso Plattner Institute
  for Digital Health at Mount Sinai, Icahn School of Medicine at Mount Sinai, New York, USA
[*]These authors contributed equally.

## Abstract

Training sophisticated machine learning (ML) models requires large datasets that are difficult or expensive to collect for many applications. If prior knowledge about system dynamics is available, mechanistic representations can be used to supplement real-world data. We present SimbaML (Simulation-Based ML), an open-source tool that unifies realistic synthetic dataset generation from ordinary differential equation-based models and the direct analysis and inclusion in ML pipelines. SimbaML conveniently enables investigating transfer learning from synthetic to real-world data, data augmentation, identifying needs for data collection, and benchmarking physics-informed ML approaches. SimbaML is available from https://pypi.org/project/simba-ml/.

## 1 Introduction and Related Work

The success of machine learning (ML) models highly depends on the quality and quantity of available data. However, collecting real-world data is costly and time-consuming. Recent advances in generative models that produce synthetic data, such as generative adversarial networks (Goodfellow et al., 2020), variational autoencoders (Wan et al., 2017), and diffusion models (Sohl-Dickstein et al., 2015), do not fully alleviate this issue as these models need to be trained on large data corpora themselves and cannot extrapolate out-of-distribution. Scientific communities have commonly developed a wealth of domain knowledge that should be leveraged (Baker et al., 2018; Alber et al., 2019). Detailed prior knowledge of interactions between modeled entities can be represented by mechanistic models, such as ordinary differential equations (ODEs). They have been used to simulate the dynamical behavior of various systems (Herty et al., 2007; Harush & Barzel, 2017; Hass et al., 2019; Baum et al., 2019). Consequently, they allow physics-informed learning via observational bias (Karniadakis et al., 2021) and generating datasets for ML benchmarks. Multiple frameworks combining data generation from mechanistic models with ML have recently been proposed (Otness et al., 2021; Takamoto et al., 2022; Hoffmann et al., 2021), see Table 1 in the Appendix for details. However, they do not allow for generating realistic data by simulating measurement errors or missing data. In addition, they are not designed for easy extension to other mechanistic models, mainly focus on benchmarking ML models, and commonly do not provide transfer learning functionalities.

We introduce SimbaML, an all-in-one framework for integrating prior knowledge of ODE models into the ML process by synthetic data augmentation. SimbaML allows for the convenient generation of realistic synthetic data by sparsifying and adding noise. Furthermore, our framework provides customizable pipelines for various ML experiments, such as identifying needs for data collection and transfer learning.

## 2 SimbaML: Features and Use Cases

SimbaML provides diverse functionalities to generate synthetic data to improve ML tasks (see Figure 1 A). As for the simulation part, we obtain time-series data by solving user-defined ODE systems. SimbaML can add different types of noise and sparsify data by randomly removing time points to

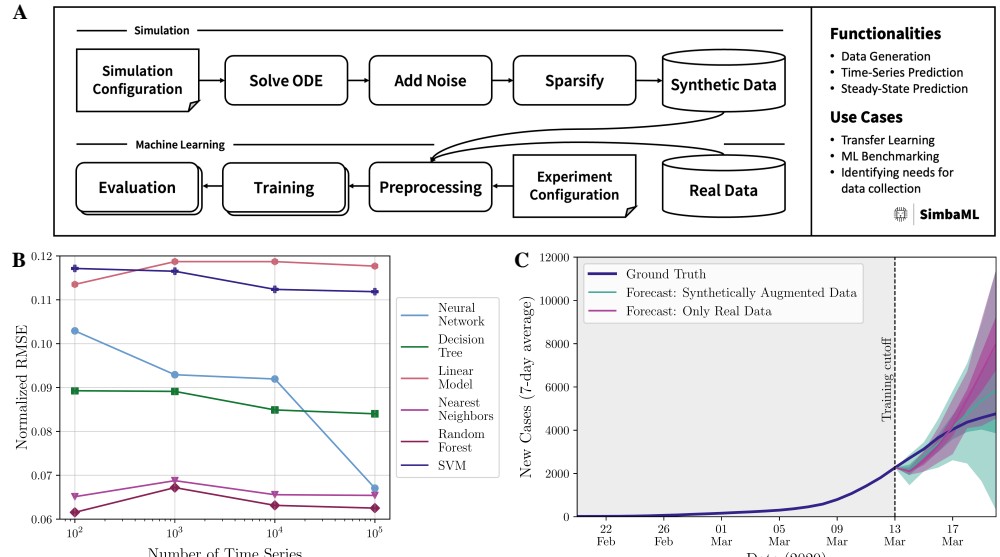

Figure 1: (A) Overview of the SimbaML framework. (B) Performance of various ML models for time-series forecasting given different amounts of synthetic time series as training data that were simulated using a biochemical pathway model. SimbaML conveniently allows end-to-end evaluation of required dataset properties, given prior knowledge of the system dynamics. (C) Comparison of two Covid-19 7-day forecasts from March 13, 2020, using a probabilistic neural network augmented with synthetic data generated by SimbaML and real-world data (we show 50% and 85% prediction intervals). This suggests that including prior knowledge using SimbaML can improve the quality of forecasts. See Appendix for details on the experiments.

make time-series data more realistic. Noise, kinetic parameters, as well as initial condition values, can be easily configured by the user. Furthermore, SimbaML offers multiple pipelines covering data pre-processing, training, and evaluation of ML models. In addition to supporting standard ML approaches, SimbaML allows for the effortless integration of any other model, for example, from Keras, PyTorch Lightning, and scikit-learn. As all pipelines can be customized based on configuration files, SimbaML enables highly automated ML experiments. Overall, SimbaML provides well-tested functionalities (100% test coverage) for various use cases, ranging from transfer learning to benchmarking and identifying needs for data collection.

We illustrate the versatility of SimbaML in two use cases. First, we use SimbaML to identify needs for data collection. We generate multiple time-series datasets with noise for a complex biochemical model of a signaling pathway (Huang & Ferrell, 1996). Based on the performance comparison for different ML models (see Figure 1 B), we can make an informed decision to use the random forest or nearest neighbor approach for smaller dataset sizes. The neural network becomes a viable candidate if sufficient data is available. Second, we use SimbaML to augment scarce real-world data (Robert Koch-Institut, 2022) for a COVID-19 time series forecasting task (see Figure 1 C). We generate noisy synthetic time series using the Susceptible-Infected-Recovered (SIR) epidemiological model (Kermack & McKendrick, 1927) with randomly sampled kinetic parameter values. Exemplary predictions on synthetically augmented training data improve compared to predictions based solely on observed data.

## 3 CONCLUSION

We present SimbaML, an all-in-one solution for generating realistic data from ODE models and leveraging them for improved ML experiments. For two exemplary use cases, we show how SimbaML effectively utilizes prior knowledge via ODEs for ML tasks in settings with limited data availability. However, we expect the degree of (in)coherence between the ODE model and the underlying real-world process to be highly influential for the success of data augmentation with SimbaML. Thus, ODE models must be chosen carefully. Future research will be conducted on the impact of different forms of noise and dataset sizes in transfer learning and data augmentation setups to further increase the applicability of our framework. As an open-source Python package, we also plan to extend SimbaML with additional functionalities, for example, out-of-the-box physics-informed and graph neural networks, to maximize the potential of mechanistic models for ML.

URM STATEMENT

The authors acknowledge that all key authors of this work meet the URM criteria of ICLR 2023 Tiny Papers Track.

ACKNOWLEDGEMENTS

This work was supported by the Bundesministerium für Wirtschaft und Klimaschutz Daten- und KI-gestütztes Frühwarnsystem zur Stabilisierung der deutschen Wirtschaft (01MK21009E) given to BYR, and by the Klaus Tschira Stiftung gGmbH (GSO/KT 25) given to KB.

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

# A    APPENDIX

## A.1    EXPERIMENTAL SETUP

We describe two example use cases that leverage SimbaML's capabilities. The code for both experiments is available at `https://github.com/DILiS-lab/SimbaML_examples`.

### A.1.1    FORECASTING IN A COMPLEX BIOCHEMICAL PATHWAY: DATA NEEDS AND DECIDING FOR AN APPROPRIATE ML MODEL

For this use case (shown in Figure 1 B), we forecast time series for a selected readout of a biochemical signaling pathway. We use SimbaML to identify needs for data collection, i.e., which amount of data is needed for training an ML model for this prediction task, and which ML model to apply.

We assume the expected measurement data to consist of time series of five time steps of a single modeled quantity. We want the ML models to be capable of predicting the subsequent three time steps of that quantity. To infuse prior knowledge on the underlying system with SimbaML, we assume the dynamics to be governed by an ODE model of the underlying biological pathway, the MAPK signaling pathway (Huang & Ferrell, 1996). The ODE system represents the dynamics of 20 interacting species. The readout quantity is the relative amount of the total maximally activated signaling molecule (this is derived as a combination of two of the modeled species and an initial condition: free plus phosphatase-bound doubly phosphorylated MAPK by total MAPK). We restrict the kinetic parameters and initial conditions to relevant ranges and sample from them to generate between $10^2$ and $10^5$ distinct time series for ML model training. We use the ODE solver LSODA (Petzold, 1983; Hindmarsh, 1983) with SimbaML to simulate 20 time steps for each time series. We render data more realistic by adding lognormal noise with SimbaML to mimic measurement noise on the simulated time series.

We use SimbaML to train and evaluate the following ML models: Neural network, decision tree, linear model, nearest neighbor, random forest, and support vector machine. We use the normalized root mean squared error (RMSE) as the performance metric.

### A.1.2    SIR-INFORMED TIME SERIES FORECASTING IN A SMALL DATA SETTING

We obtain daily reports on new cases of COVID-19 in Germany during the initial wave of the pandemic in early 2020 from Robert Koch-Institut (2022) and preprocess the data using a 7-day-moving average.

Using SimbaML, we generate 100 synthetic pandemic incidence time series with an SIR - ODE compartment model (Kermack & McKendrick, 1927). We initialize the susceptible compartment $S$ with the estimated population of Germany $N$ at the end of 2019 (83,166,711) subtracted by 100 initially infected individuals $I$, and zero recovered individuals. Within SimbaML we sample the transmission rate ($\beta$) and recovery rate ($\gamma$) parameters from continuous uniform distributions with support $(0.32, 0.35)$ and $(0.123, 0.125)$, respectively. We use SimbaML's additive Gaussian noise to mimic measurement errors synthetically. Alongside the standard SIR model, we simulate the cumulative new cases $C_\Sigma$ with its derivative:

$$\frac{dC_\Sigma}{dt} = \frac{\beta S I}{N}$$

and calculate the finite difference to attain the new cases for each timepoint $t$:

$$C(t) = C_\Sigma(t) - C_\Sigma(t - 1).$$

We train a neural network that predicts the parameters of a Student's t-distribution for the new case number using seven days as input and forecasting horizon, respectively. We use a feed-forward architecture as implemented in the Python library GluonTS (Alexandrov et al., 2020) with two layers of 20 neurons each. We define a training cut-off in the early stage of the pandemic, before the peak of the first wave, and compare the prediction of a model trained solely on the observed time series to a model trained additionally on all synthetically generated time series (see Figure 1 C).

## A.2    RELATED WORK

We reviewed relevant publications that use or enable data generation with mechanistic modeling and time series prediction. We highlight differences from SimbaML in Table 1.

Table 1: Comparison of SimbaML to related work in five areas. First, on the data generation capabilities, adding noise and sparsification of the generated data for realistic real-world data. Second, the supported prediction tasks of time series as well as steady states. Third, on the use cases, analyzing for both benchmark and transfer learning capabilities. Fourth, on the software aspect of extendability, the scope of the use-case, and software licensing. Fifth, we analyzed the functionalities not focused on limited data scenarios, but that are likewise related to the combination of machine learning with data generated from mechanical models: learning of ODEs as functions from data and solving of computationally expensive mechanistic models. 1: Takamoto et al. (2022) (PDEBench), 2: Hoffmann et al. (2021) (Deeptime), 3: Otness et al. (2021), 4: Costello & Martin (2018), 5: Raissi et al. (2020) (HFM), 6: Bar-Sinai et al. (2019), 7: Raissi et al. (2019)

| Related Work | SimbaML | Frameworks | | | Use Cases | | Related | |
|---|---|---|---|---|---|---|---|---|
| | | 1 | 2 | 3 | 4 | 5 | 6 | 7 |
| Generate Data | x | x | x | x | x | x | x | x |
| Noise and Sparsification | x | x | | | | | | |
| Predict Time Series | x | x | x | x | x | x | x | x |
| Predict Steady State | x | x | | | | | | |
| Benchmarking of ML | x | x | x | x | x | | | x |
| Transfer Learning | x | | | | x | x | | |
| Extendable | x | | x | x | | | | |
| Generic Use-Case | x | x | x | x | | | | |
| Open Source | x | x | x | x | | x | x | x |
| Learning ODE/PDE | | | | | | | x | x |
| Solving ODE/PDE With ML | | | | x | | x | | x |

