# OpenReview forum: "SimbaML: Connecting Mechanistic Models and Machine Learning with Augmented Data"
_ICLR.cc/2023/TinyPapers — Submitted to Tiny Papers @ ICLR 2023_

### Official Review · Reviewer_BGjs · 2023-03-24

**Confidence:** 4

**Summary Of Contributions:**

The SimbaML tool presented in this paper unifies realistic synthetic dataset generation from ordinary differential equation-based models and the direct analysis and inclusion in ML pipelines. It provides a solution for training sophisticated machine learning models that require large datasets that are difficult or expensive to collect for many applications.

**Rating:**

High Potential (HP): a submission which meets the reviewing criteria and has potential to make an impact on the field

**Strengths And Weaknesses:**

Strengths:
1. SimbaML can supplement real-world data with prior knowledge about system dynamics, which can improve the accuracy and efficiency of machine learning models.
2. The tool is open-source and written in Python, making it accessible to a wide range of users.

Weaknesses:
1. SimbaML requires prior knowledge about system dynamics, which may not always be available or accurate.
2. The scalability of the tool may be limited by the computational resources required to generate synthetic datasets from ordinary differential equation-based models.
3. There is a risk of overfitting to synthetic data, which may not accurately represent real-world scenarios.



**Suggested Changes:**

1. Expand the tool's capabilities beyond ordinary differential equation-based models to include other types of mechanistic models.
2. Address scalability concerns by optimizing the computational resources required to generate synthetic datasets and analyzing large datasets efficiently.
3. Further explore the potential for overfitting and develop methods to mitigate this risk, such as incorporating real-world data into the synthetic dataset generation process.

---

> ### Author Response · Authors · 2023-06-01
> **Guidelines on best usage and extending our CPU-parallelizable SimbaML with further models are planned**
>
> We thank the reviewer for their comments. We would like to add that indeed, SimbaML works best when prior knowledge (in form of an ODE model) on the underlying system exists, but that SimbaML is modularly structured – even if no simulations based on ODEs are performed, its interface can be used equally well just for ML-based predictions to make related analyses easily possible.
> SimbaML is being actively developed, and we agree with the reviewer that the support for more mechanistic model types would broaden the scope of SimbaML considerably. In principle, an extension to other types of equation-based models is possible due to SimbaMLs modular implementation approach and is planned to be addressed in one of the next versions of the tool.
> As the reviewer mentions, runtime for synthetic data generation might indeed be an issue for very complex ODE models. SimbaML’s data generation process is by design highly parallelizable (on CPU) as additional synthetic data samples can be generated on each separate CPU. Also, in the main target application domain of (molecular) biology, real-world data generation is usually very costly, time-consuming, and, in case of animal or human samples, sometimes unethical, and so simulating synthetic data from ODE models is in all cases comparably low in effort.
> It is important to mention that SimbaML in its current form does not offer automated means to prevent overfitting on synthetic data or other detrimental effects of wrong prior knowledge in the form of unsuited ODE models (we added a statement on that into the conclusion section in the revised paper). However, we are currently developing practical guidelines for hyperparameters such as settings of noise or the effect of ratios of synthetic vs. real-world data for prediction pipelines. As these experiments need to be carefully designed and comprehensive enough to allow for valid conclusions, we chose not to include them in this work and will provide them in the accompanying github repository instead.

---

### Official Review · Reviewer_uxGY · 2023-03-30

**Confidence:** 4

**Summary Of Contributions:**

The submission presents SimbaML, an open-source tool that generates realistic synthetic datasets from ordinary differential equation-based models to supplement real-world data in machine learning models. SimbaML allows for investigating transfer learning, data augmentation, identifying the need for data collection, and benchmarking physics-informed ML approaches. SimbaML provides diverse functionalities covering data pre-processing, training, and evaluation of ML models.

**Rating:**

High Potential (HP): a submission which meets the reviewing criteria and has potential to make an impact on the field

**Strengths And Weaknesses:**

## Strengths

- The paper addresses a relevant and important problem: the need for large and diverse datasets for training machine learning models.
- The proposed framework, SimbaML, seems to be a comprehensive solution that integrates mechanistic models and machine learning for dataset generation and analysis.
- The paper includes two use cases that demonstrate the potential of SimbaML in identifying needs for data collection and augmenting scarce real-world data.
- I appreciate that the authors have been abundantly clear throughout the paper and in the appendix. They clearly communicate the goals of SimbaML as well as contrast it with other frameworks and tools.
- The work provides high-quality reproducible code.

## Weaknesses

- The paper assumes that the mechanistic models used to simulate the dynamical behavior of various systems are accurate and unbiased. However, these models are constructed based on assumptions and simplifications of the underlying physics, and they may not fully capture the complexity of real-world systems. SimbaML provides no way to do so and does not study how this impacts the functionality of their work.

**Suggested Changes:**


Overall, the submission is well-structured, concise, and clearly presents the SimbaML framework for integrating prior knowledge of ODE models into the ML process by synthetic data augmentation. However, the authors may consider making the following changes to improve the submission:

- The authors may consider adding more details about the SimbaML framework's technical aspects, such as the data pre-processing, training, and evaluation pipelines, to provide a better understanding of how the framework works.
- In Figure 1B, the figure could be more descriptive to provide a better understanding of the size of the dataset as well as the dataset used for each experiment.
- I have made sure to not actively look for the authors of this work, however, the code repository includes a link to PyPI which unfortunately reveals the authors. I have made sure to not search up the names and since this does not seem intentional as per recommendation this will be ignored.

---

> ### Author Response · Authors · 2023-06-01
> **Choose your ODE models with care, and find SimbaML’s comprehensive documentation at https://simbaml.readthedocs.io/**
>
> We appreciate the reviewer’s comments and are happy that they liked our approach. Indeed, the quality of the underlying ODE model and the impact on the usefulness of the synthetic data generated with SimbaML is an important issue that needs further consideration and deserves a dedicated set of experiments as part of future work. We added a corresponding statement into the discussion section of our revised paper version.
>
> “However, we expect that the degree of (in)coherence between the ODE model and the underlying real-world process will be highly influential for the success of data augmentation with SimbaML. Thus, ODE models must be chosen carefully.”
>
> Also, we invite everyone to visit the comprehensive documentation to SimbaML at https://simbaml.readthedocs.io/ in that all its technical aspects are described in-depth.
> In our revised version of the manuscript, we also adapted the caption to Fig. 1B in order to make the setup clearer for the reader. Please also refer to the Appendix for more details on our examples.

---

### Meta-Review · Area_Chair_w5XF · 2023-04-08

**Recommendation:** Invite to present (notable)
**Confidence:** 4

**Metareview:**

The submitted paper presents SimbaML, an open-source tool that generates realistic synthetic datasets from ordinary differential equation-based models to supplement real-world data in machine learning models. SimbaML offers diverse functionalities covering data pre-processing, training, and evaluation of ML models, and can investigate transfer learning, and data augmentation, identify the need for data collection, and benchmark physics-informed ML approaches. This toolkit can serve a wide of applications and have a potential impact on this area.

While the paper addresses an important and relevant problem, the paper assumes the accuracy and bias of the underlying physics in the models used to simulate dynamical behavior without studying the impact on the functionality of SimbaML and the risk of overfitting to synthetic data. The authors are encouraged to further investigate this issue.

**Summary:**

The submission presents SimbaML, an open-source tool that generates realistic synthetic datasets from ordinary differential equation-based models to supplement real-world data in machine learning models. SimbaML offers diverse functionalities covering data pre-processing, training, and evaluation of ML models, and can investigate transfer learning, data augmentation, identify the need for data collection, and benchmark physics-informed ML approaches.

**Reason For Not Giving A Higher Recommendation:**

N/A

**Reason For Not Giving A Lower Recommendation:**

The submission addresses an important and relevant problem in the field of machine learning, and the proposed framework, SimbaML, offers comprehensive solutions to generate synthetic datasets and analyze machine learning models. The paper provides high-quality reproducible code, and the authors have been clear throughout the paper and the appendix.

---

> ### Author Response · Authors · 2023-06-01
> **The choice of the ODE model and practical guidelines for SimbaML**
>
> We appreciate the feedback of all reviewers. We have included comments on potential issues with incoherence between the ODE model and the underlying real-world process that should be predicted with synthetically augmented data. Also, we are actively working on practical guidelines for hyperparameters such as settings of noise or the effect of ratios of synthetic vs. real-world data for prediction pipelines. We will provide them on SimbaML’s github repository so it can be easily extended and does not face any restrictions in space and format. We thank the reviewers for giving us the great opportunity to present our work as a talk and paper at ICLR 2023!

---

### Decision · Program_Chairs · 2023-04-09

Invite to present (notable)